

# Application of the adjoint approach to optimise the initial conditions of a turbidity current

Samuel D. Parkinson[1], Simon W. Funke[2], Jon Hill[3], Matthew D. Piggott[1,4], and
Peter A. Allison[1]

[1]Department of Earth Science and Engineering, Imperial College London, UK
[2]Center for Biomedical Computing, Simula Research Laboratory, Oslo, Norway
[3]Environment Department, University of York, UK
[4]Grantham Institute – Climate Change and the Environment, Imperial College London, UK

*Correspondence to:* Simon W. Funke (simon@simula.no)

**Abstract.**

Turbidity currents are one of the main drivers for sediment transport from the continental shelf to the deep ocean. The resulting sediment deposits can reach hundreds of kilometres into the ocean. Computer models that simulate turbidity currents and the resulting sediment deposit can help to
understand their general behaviour. However, in order to recreate real-world scenarios, the challenge is to find the turbidity current parameters that reproduce the observations of sediment deposits.

This paper demonstrates a solution to the inverse sediment transportation problem: for a known sedimentary deposit, the developed model reconstructs details about the turbidity current that produced these deposits. The reconstruction is constrained here by a shallow water sediment-laden den-
sity current model, which is discretised by the finite element method and an adaptive time-stepping scheme. The model is differentiated using the adjoint approach and an efficient gradient-based optimisation method is applied to identify turbidity parameters which minimise the misfit between modelled and observed field sediment deposits. The capabilities of this approach are demonstrated using measurements taken in the Miocene-age Marnoso Arenacea Formation (Italy). We find that whilst
the model cannot match the deposit exactly due to limitations in the physical processes simulated, it provides valuable insights into the depositional processes and represents a significant advance in our toolset for interpreting turbidity current deposits.

## 1 Introduction

Turbidity currents are density currents driven by sediment particles that are suspended by turbulence
in the containing fluid (Lowe, 1979). They occur frequently throughout the Earth's oceans and are one of the main processes by which sediment is moved from the continental shelf to the deep ocean. The largest turbidity currents can involve several hundred cubic-kilometres of sediment (Talling



et al., 2007c), and can travel for hundreds of kilometres across the sea bed at speeds of tens of metres per second (Heezen and Ewing, 1952).

The vast majority of available data for turbidity currents is contained in the sedimentary deposits that they leave behind. Significant effort is spent in attempting to diagnose details about the turbidity current that produced them. Talling et al. (2007a) and Talling et al. (2012) describe the current theories for how deposits found in the field are formed. The experimental evidence cannot yet validate all of these theories. Computer models, along with laboratory experiments, have been useful tools

in improving our understanding of the dynamics of turbidity currents (Talling et al., 2007a; Kneller and Buckee, 2000; Parkinson et al., 2014). However, computer models have not often been directly applied to recreating deposits found in the field despite their capacity to do so. They are generally applied on idealised cases to understand a specific physical mechanism. It is useful to directly apply models in attempts to recreate real-world deposits (Fukushima et al., 1985; Huang et al., 2009;

Doyle et al., 2010) but this requires good knowledge of the initial and boundary conditions, and accurate estimates of values for other controlling model parameters which are often hard to determine (Talling et al., 2007a).

The task of obtaining a set of model input parameters based upon a desired model output represents an inverse problem. It can also be interpreted as an optimisation problem where model param-

eters are sought to minimise the misfit between the deposit profile generated by the model, and a target deposit profile, which is produced from measurements taken in the field.

In this paper a shallow water model is used to simulate turbidity currents. The shallow water equations are a set of partial differential equations (PDEs). Optimisation of PDE-based models occurs throughout science and engineering, and is already applied for instance in ocean science (Menemen-

lis et al., 2008), renewable energy (Funke et al., 2014) and design problems (Giles and Pierce, 2000). In addition, there is a growing interest in applying inverse modelling techniques to the modelling of turbidity currents (Naruse, 2013; Rowley, 2013).

PDE models of turbidity currents require the definition of initial and boundary conditions. In the simplest case this could involve the definition of a static lock-release laboratory configuration

with uniform sediment depth and a single, uniform sediment grain size. Such a simple configuration would at least require the definition of the initial depth of the current, concentration of sediment in the fluid, ratio of initial depth to length, and parameters controlling the particle settling velocity and flow front speed. More realistic initial conditions would be an inflow condition with time-varying depth, velocity, and concentrations of a wide range of sediment grain sizes, along with information

defining the topography of the bed, its composition, and parameterisations governing bed erosion rates, flow rheology, and bedload transport. As the choice of boundary and initial conditions, and model complexity increase, the range of deposit shapes that can be generated by the model increase such that it is capable of better recreating a range of deposits found in the field. However, with this



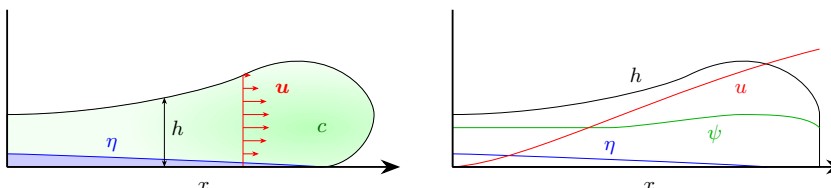

**Figure 1.** Schematic representation of a dense gravity current (left), and a corresponding depth-averaged shallow water approximation (right), shows current height $h$, volume fraction $c$, and depth-averaged volume fraction $\psi$, velocity $\boldsymbol{u}$, and forward component of the depth-averaged velocity $u$, and deposit depth $\eta$ .

added complexity, the parameter space grows and manual tuning of parameters becomes a greater
challenge.

This paper presents a shallow water sediment-laden density current model that uses a novel finite element mixed discontinuous Galerkin function space, with adaptive time-stepping (Section 2). The model implementation is verified through comparison with analytical solutions and convergence analyses (Section 2.5). The model is then differentiated using the adjoint method, which is an effi-
cient way of computing the sensitivity of a model output to many input parameters (Section 3). This enables the use of fast converging gradient-based optimisation techniques. Finally, a gradient-based optimisation technique is applied to minimise data misfit between the modelled sediment deposit and field measurements taken in the Miocene-age Marnoso Arenacea Formation (Section 4). To the best of the authors' knowledge this paper represents the first published work with optimisation applied to
turbidity currents and demonstrates the usefulness of these techniques for interpreting sedimentary successions that have been deposited by turbidity currents.

## 2   Model

Shallow water models solve the Navier-Stokes equations in depth-averaged form (Figure 1). They are a valid approximation when the horizontal length scale, or length of the current, is much larger
than the vertical length scale, the height of the current. This is the case for sediment-laden density currents for all times a short period after an initial release. In this case the vertical pressure gradients are in near hydrostatic balance. Sediment in the current is assumed to be well mixed by the turbulence in the flow such that there is a vertically-uniform sediment distribution.

Shallow water sediment-laden density current models come in a variety of forms. Parker et al.
(1986) proposed the 'four-equation' model. This is a complex model which accounts for entrainment of sediment from the bed, and entrainment of ambient fluid into the flow. It has an extra equation for the internal kinetic energy of the flow, which is translated into potential energy through these mixing processes. A drag force is applied along the length of the current which takes account of the viscous forces retarding the flow motion at the base, and top of the flow. This model has been applied





to the modelling of large-scale turbidity currents (Fukushima et al., 1985; Huang et al., 2009). It is dependent upon the selection of numerous governing parameters and hence is a good use case for inverse-modelling. A similar, but slightly simplified model was used by Doyle et al. (2008, 2010) for modelling the plume of a dense pyroclastic basal flow. This model also included a dense underflow. This has been applied in direct comparison with field measurements (Doyle et al., 2010).

Bonnecaze et al. (1993) proposed one and two-layer sediment-laden shallow water density current models. The two-layer model includes equations for the motion of the ambient fluid through which the density current is propagating, which is important where the ambient fluid depth is similar to the initial current depth. The one and two-layer models presented by Bonnecaze et al. (1993) use a coordinate system that adapts relative to the length of the flow. The moving coordinate system allows

the speed to be prescribed at the front of the current. This speed can be well approximated using the Froude number, the height, and the volume fraction of sediment in the current. This is a good approach as the speed of the front of a gravity current is governed by dynamics that cannot be resolved by a vertically averaged model. The moving coordinate system also results in a discretisation that scales with the horizontal length scale of the flow. This is beneficial for capturing the important flow

features. This model has been used extensively in understanding turbidity current flow characteristics (Harris et al., 2001; Hogg, 2006), including the effects of modelling polydisperse suspensions (Harris et al., 2002; Garcia, 1994) and the effects of external flow (Hallworth et al., 1998). The model used in this paper is based upon the single layer shallow water model of Bonnecaze et al. (1993).

### 2.1 Governing equations

The equations governing the current column height, $h$, and vertically integrated momentum, $q = uh$, with $u$ being the depth-averaged current velocity, are described in non-dimensionalised conservative form as

$$\frac{\partial h}{\partial t} + \frac{\partial q}{\partial x} = 0 \,, \tag{1}$$

$$\frac{\partial q}{\partial t} + \frac{\partial}{\partial x}\left(\frac{q^2}{h} + \frac{\varphi h}{2}\right) = 0 \,, \tag{2}$$

with boundary conditions

$$q = 0 \ \text{at} \ x = 0 \,, \tag{3}$$

$$q = \dot{x}_N h \ \text{at} \ x = x_N(t) \,, \tag{4}$$

given that

$$\dot{x}_N = Fr\varphi^{1/2} \ \text{at} \ x = x_N(t) \,, \tag{5}$$

where $x_N$ is the location of the front of the current, $\dot{x}_N$ is the velocity of the front of the current, $Fr$ is the Froude number, and $\varphi = \psi h$ is the vertically-integrated volume fraction of sediment where $\psi$ is the depth-averaged volume fraction of sediment within the flow. Through experimentation, the



Froude number for a density current, with head height $< 0.075$ of the total water depth, has been found to be 1.19 (Huppert and Simpson, 1980). The evolution of $\varphi$ is described using

$$\frac{\partial \varphi}{\partial t} + \frac{\partial}{\partial x}\left(\frac{q\varphi}{h}\right) = -\beta\frac{\varphi}{h} \, , \tag{6}$$

where $\beta$ is a constant particle settling parameter. Hence the gravitational forcing term in (2) (the last term on the left hand side) changes with time as $\varphi$ is advected and settles out of the column.

This single layer model ignores the effect of the motion of the overlying fluid on the current. This approximation is valid for flows where the maximum column height is significantly less than the depth of the ambient fluid (Bonnecaze et al., 1993; Hogg, 2006). Viscous forces are also ignored. For high Reynolds number flows the viscous forces will be negligible in relation to the buoyancy forces. Bonnecaze et al. (1993) found that this was valid while the Reynolds number was greater than $\mathcal{O}(1)$.

The amount of deposited sediment, $\eta$, is also recorded and is calculated using

$$\frac{\partial \eta}{\partial t} = \beta\frac{\varphi}{h} \, . \tag{7}$$

The model is non-dimensionalised with the length, time, and velocity scales $h_0$, $(h_0/g_0')^{1/2}$, and $(h_0 g_0')^{1/2}$ respectively. Here, $h_0$ is the dimensional depth of the initial sediment release, $g_0' = \psi_0 g (\rho_p - \rho_a)/\rho_a$ is the initial reduced gravity of the current, $\rho_p$ is the sediment particle density, $\rho_a$ is the ambient fluid density, which is assumed to equal the interstitial fluid density, $\psi_0$ is the initial volume fraction of sediment, and $g$ is the acceleration due to gravity. Finally, the volume fraction is scaled such that $\int_0^{x_N(0)} \varphi \, dx = 1$.

Following Bonnecaze et al. (1993), a coordinate transformation from $(x, t)$ to $(y, \tau)$ is applied where $y = x/x_N(t)$ and $t = \tau$. This is a convenient form for the equations as the front of the current is always at the right hand boundary of a fixed computational domain, and hence the boundary condition at the front of the flow is applied at the right-hand side of the domain. The transformed derivatives are given by

$$\frac{\partial}{\partial t} = \frac{\partial}{\partial \tau} - \frac{y\dot{x}_N}{x_N}\frac{\partial}{\partial y} \, , \tag{8}$$

$$\frac{\partial}{\partial x} = \frac{1}{x_N}\frac{\partial}{\partial y} \, . \tag{9}$$





Applying this coordinate transformation, but keeping $t$ in place of $\tau$ for notational simplicity

following Bonnecaze et al. (1993), produces the system of equations:

$$\frac{\partial h}{\partial t} = \frac{1}{x_N} \left( y \dot{x}_N \frac{\partial h}{\partial y} - \frac{\partial q}{\partial y} \right) , \tag{10}$$

$$\frac{\partial q}{\partial t} = \frac{1}{x_N} \left( y \dot{x}_N \frac{\partial q}{\partial y} - \frac{\partial}{\partial y} \left( \frac{q^2}{h} + \frac{\varphi h}{2} \right) \right) , \tag{11}$$

$$\frac{\partial \varphi}{\partial t} = \frac{1}{x_N} \left( y \dot{x}_N \frac{\partial \varphi}{\partial y} - \frac{\partial}{\partial y} \left( \frac{q \varphi}{h} \right) \right) - \beta \frac{\varphi}{h} , \tag{12}$$

$$\frac{\partial \eta}{\partial t} = \frac{1}{x_N} \left( y \dot{x}_N \frac{\partial \eta}{\partial y} \right) + \beta \frac{\varphi}{h} , \tag{13}$$

$$\frac{\partial x_N}{\partial t} = \dot{x}_N , \tag{14}$$

with boundary conditions

$$q = 0 \ \text{at} \ y = 0 , \tag{15}$$

$$q = \dot{x}_N h \ \text{at} \ y = 1 , \tag{16}$$

$$\eta = 0 \ \text{at} \ y = 1 , \tag{17}$$

given that

$$\dot{x}_N = Fr \varphi^{1/2} \ \text{at} \ y = 1 . \tag{18}$$

Note that an equation (14) for $x_N$ has been introduced to close the system.

It is now shown that the boundary conditions (15) to (17) are sufficient to uniquely solve this system. Equations (10) to (13) are a hyperbolic system of PDEs. For such a system to be well posed

there must be a boundary conditions for each inwardly propagating characteristic. This system of equations has four characteristic velocities

$$\frac{\mathrm{d}y}{\mathrm{d}t} = c_{\pm} := \frac{1}{x_N} \left( u - y \dot{x}_N \pm \varphi^{1/2} \right) , \tag{19}$$

$$\frac{\mathrm{d}y}{\mathrm{d}t} = c := \frac{1}{x_N} \left( u - y \dot{x}_N \right) , \tag{20}$$

$$\frac{\mathrm{d}y}{\mathrm{d}t} = c_{\eta} := -\frac{1}{x_N} y \dot{x}_N . \tag{21}$$

These are obtained using the method of characteristics. $c_{\pm}$ is the characteristic velocity of waves in shallow water, $c$ is the advection velocity of sediment, and $c_{\eta}$ is the advection velocity of deposited sediment which is advected away from the current head as the domain length increases.

Due to the boundary conditions on momentum the following is true, $u = q/h = 0$ at $y = 0$ and $u = q/h = \dot{x}_N$ at $y = 1$. Hence, $c = 0$ at both $y = 0$ and $y = 1$. Therefore there are three inwardly

propagating characteristics, $c_+ = \varphi^{1/2}/x_N$ at $y = 0$, $c_- = -\varphi^{1/2}/x_N$ at $y = 1$ and $c_{\eta} = -y \dot{x}_N/x_N$ at $y = 1$. Hence three boundary conditions are required for the problem to be well posed such that the three boundary conditions (15) to (17) are exactly what is required.





### 2.2 Discretisation and numerical method

As the cell size grows throughout the simulation, it is possible to use a much larger timestep at the
end of the simulation than at the start of the simulation. To exploit this property an adaptive time-
stepping scheme is used in this model. A new time-dependent variable is introduced, $\Delta t$, which will
vary according to a CFL-criteria, $C$, based upon a velocity scale $\dot{x}_N$ and the mesh element size such
that

$$\Delta t = C\,\frac{\Delta_x}{\dot{x}_N} = C\,\frac{x_N \Delta_y}{\dot{x}_N}\;, \tag{22}$$

where $\Delta_x$ is the mesh element size in $x$ and $\Delta_y$ is the mesh element size in the transformed coordi-
nate system $y$.

The time-dependent model variables are defined as a vector

$$U = [\,h,\,q,\,\varphi,\,\eta,\,x_N,\,\dot{x}_N,\,\Delta t\,]^T\;. \tag{23}$$

The system is discretised in time using a second-order explicit Runge-Kutta time discretisa-
tion (Cockburn and Shu, 2001). An implicit term is added to the semi-discrete system in order
to solve for the diagnostic variables, $\dot{x}_N$ and $\Delta t$. With $U^n$ as the solution at the beginning of the
timestep, $U^{n+1}$ as the solution at the end of the timestep, and $U^{(0)}$, $U^{(1)}$ and $U^{(2)}$ being intermediate
values, the system of equations, discretised in time can be written as

$$\begin{aligned}
U^{(0)} &= U^n\,, \\
U^{(1)} &= A(U^{(0)}) + L(U^{(0)}) + K(U^{(1)})\,, \\
U^{(2)} &= A(U^{(1)}) + L(U^{(1)}) + K(U^{(2)})\,, \\
U^{n+1} &= \frac{1}{2}U^{(0)} + \frac{1}{2}U^{(2)}\,,
\end{aligned} \tag{24}$$

where

$$A(U) = [\,h,\,q,\,\varphi,\,\eta,\,x_N,\,0,\,0\,]^T\;, \tag{25}$$

$$L(U) = \Delta t\left(\frac{1}{x_N}\left(y\dot{x}_N\frac{\partial f_1(U)}{\partial x} - \frac{\partial f_2(U)}{\partial x}\right) + f_3(U)\right)\;, \tag{26}$$

$$f_1 = [\,h,\,q,\,\varphi,\,\eta,\,0,\,0,\,0]^T\;, \tag{27}$$

$$f_2 = \left[\,q,\,\frac{q^2}{h} + \frac{\varphi h}{2},\,\frac{q\varphi}{h},\,0,\,0,\,0,\,0\,\right]^T\;, \tag{28}$$

$$f_3 = \left[\,0,\,0,\,-\beta\frac{\varphi}{h},\,\beta\frac{\varphi}{h},\,0,\,0,\,0\,\right]^T\;. \tag{29}$$

$A(U)$ is non-zero where there is a time derivative term. $L(U)$ is the explicit right hand side terms
multiplied by $\Delta t$. Note that $K(U)$ contains the implicit right hand side terms. $K(U)$ can only be
easily described well in weak form, so this is defined later.





The spatial weak form of the semi-discrete system (24) is obtained by multiplying by a test func-
tion, $\Psi$, and integrating over the domain, $\Omega$. This gives, for all $\Psi$ in an appropriately chosen test
space

$$\int_\Omega \Psi \cdot U^{(0)} \, d\Omega = \int_\Omega \Psi \cdot U^n \, d\Omega \,,$$

$$\int_\Omega \Psi \cdot U^{(1)} \, d\Omega = \int_\Omega \Psi \cdot A(U^{(0)}) \, d\Omega +$$

$$\int_\Omega \Psi \cdot L(U^{(0)}) \, d\Omega + \int_\Omega \Psi \cdot K(U^{(1)}) \, d\Omega \,,$$

$$\int_\Omega \Psi \cdot U^{(2)} \, d\Omega = \int_\Omega \Psi \cdot A(U^{(1)}) \, d\Omega + \tag{30}$$

$$\int_\Omega \Psi \cdot L(U^{(1)}) \, d\Omega + \int_\Omega \Psi \cdot K(U^{(2)}) \, d\Omega \,,$$

$$\int_\Omega \Psi \cdot U^{n+1} \, d\Omega = \frac{1}{2} \int_\Omega \Psi \cdot \left( U^n + U^{(2)} \right) \, d\Omega \,.$$

Piecewise-linear discontinuous Galerkin (DG) elements are used to discretise the spatially varying
state variables. Thus the spatial and temporal discretisations both have second-order accuracy. DG
element types are known to be particularly suitable for advection dominated problems (Peraire and
Persson, 2008). They are good at preserving discontinuities as they produce stable discretisations
without the need for diffusive stabilisation strategies such as streamline-upwinding (Peraire and
Persson, 2008). These are important features in shallow water particle-laden density current models.

In order to construct a DG formulation a regular partition, $\mathcal{T}_h = \{e\}$, of $\Omega$ into non-overlapping
sub-domains $\Omega_e \in \Omega$ with boundaries $\partial\Omega_e$ is considered. The piecewise-linear DG function space
is denoted $\mathbf{DG}_1$. For this function space, piecewise-linear test functions with no global continuity
requirement are considered, i.e. functions that have the potential to be double-valued on $\partial\Omega_e$. $x_N$,
$\dot{x}_N$ and $\Delta t$ are defined on a function space, $\mathbf{R}$, which is constant throughout the spatial domain.
Therefore, the vector of model unknowns $U$ is defined on a mixed function space, $\mathcal{V}_h = \mathbf{DG}_1{}^4 \times$
$\mathbf{R}^3$. The test function in the mixed function space is denoted with $\Psi_h \in \mathcal{V}_h$, and the discretised
approximation of the state variable with $U_h \in \mathcal{V}_h$.

Notice that $L(U)$ contains derivatives of discontinuous functions. Its undiscretised weak form is

$$\int_\Omega \Psi \cdot L(U) \, d\Omega = \Delta t \int_\Omega \Psi \quad \cdot \left( \frac{1}{x_N} \left( y \dot{x}_N \frac{\partial f_1(U)}{\partial x} - \frac{\partial f_2(U)}{\partial x} \right) + f_3(U) \right) \, d\Omega \,. \tag{31}$$

The discretised DG formulation of (31) is then

$$\sum_{e \in \mathcal{T}_h} \int_{\Omega_e} \Psi_h \cdot L(U_h) \, d\Omega =$$


$$\Delta t \sum_{e \in \mathcal{T}_h} \int_{\Omega_e} \Psi_h \cdot \left( \frac{1}{x_N} \left( y \dot{x}_N \frac{\partial f_1(U_h)}{\partial x} - \frac{\partial f_2(U_h)}{\partial x} \right) + \Psi_h \, f_3(U_h) \right) \, d\Omega \,. \tag{32}$$





Integrating the gradient terms by parts and rearranging slightly yields

$$\sum_{e \in \mathcal{T}_h} \int_{\Omega_e} \Psi_h \cdot \frac{x_N}{\Delta t} L(U_h) \, d\Omega =$$

$$- \sum_{e \in \mathcal{T}_h} \int_{\Omega_e} \frac{\partial}{\partial x} (\Psi_h \, y \, \dot{x}_N) \cdot f_1(U_h) \, d\Omega$$

$$+ \sum_{e \in \mathcal{T}_h} \int_{\partial\Omega_e} \widehat{\Psi_h} \cdot y \, \dot{x}_N \, \widehat{f}_1(U_h) \, \widehat{n} \, d\sigma + \sum_{e \in \mathcal{T}_h} \int_{\Omega_e} \frac{\partial \Psi_h}{\partial x} \cdot f_2(U_h) \, d\Omega \tag{33}$$

$$- \sum_{e \in \mathcal{T}_h} \int_{\partial\Omega_e} \widehat{\Psi_h} \cdot \widehat{f}_2(U_h) \, \widehat{n} \, d\sigma + \sum_{e \in \mathcal{T}_h} \int_{\Omega_e} \Psi_h \cdot x_N f_3(U_h) \, d\Omega \, ,$$

where $\widehat{\cdot}$ indicates that the function is double-valued and special attention is required. The various summations can now be rewritten as integrals over the entire domain $\Omega$, all element interfaces $\Sigma_h$, and the domain boundaries $\Gamma_h$. Note that

$$\sum_{e \in \mathcal{T}_h} \int_{\partial\Omega_e} \Psi_h \cdot \widehat{U_h} \, d\sigma \equiv \int_{\Sigma_h} \Psi_h \cdot \widehat{U_h} \, d\sigma + \int_{\Gamma_h} \Psi_h \cdot U_0 \, d\sigma \, , \tag{34}$$

and

$$\sum_{e \in \mathcal{T}_h} \int_{\Omega_e} \Psi_h \cdot U_h \, d\Omega \equiv \int_{\Omega} \Psi_h \cdot U_h \, d\Omega \, . \tag{35}$$

Additionally, note that within domain boundary integrals the $\widehat{\cdot}$ notation is dropped as the function is single-valued at this location, and also that $U_h$ is replaced with $U_0$ which is either the boundary value if a Dirichlet boundary condition is present, or the function value at the boundary if it is not. Note that in the case of the boundary condition for $q$ at $y = 1$, $U_0$ is still a function of $U_h$. Applying (34) and (35) to (33) yields

$$\int_{\Omega} \Psi_h \cdot \frac{x_N}{\Delta t} L(U_h) \, d\Omega =$$

$$- \int_{\Omega} \frac{\partial}{\partial x} (\Psi_h \, y \, \dot{x}_N) \cdot f_1(U_h) \, d\Omega + \int_{\Sigma_h} \widehat{\Psi_h} \cdot y \, \dot{x}_N \, \widehat{f}_1(U_h) \, \widehat{n} \, d\sigma$$

$$+ \int_{\Gamma_h} \Psi_h \cdot y \, \dot{x}_N \, f_1(U_0) \, n \, d\sigma$$

$$+ \int_{\Omega} \frac{\partial \Psi_h}{\partial x} \cdot f_2(U_h) \, d\Omega - \int_{\Sigma_h} \widehat{\Psi_h} \cdot \widehat{f}_2(U_h) \, \widehat{n} \, d\sigma \tag{36}$$

$$- \int_{\Gamma_h} \Psi_h \cdot f_2(U_0) \, n \, d\sigma$$

$$+ \int_{\Omega} \Psi_h \cdot x_N f_3(U_h) \, d\Omega \, .$$

A choice of flux term must be made to handle the double valued terms. This will involve some coupling between the elements on either side of the interface. An upwind flux is used for the advection





term, $\widehat{f}_1$, and based upon experience an average flux works well for $\widehat{f}_2$. This gives

$$
\begin{aligned}
\int_\Omega \Psi_h \cdot \frac{x_N}{\Delta t} L(U_h) \, d\Omega = {} & \\
& - \int_\Omega \frac{\partial}{\partial x} \left( \Psi_h \, y \, \dot{x}_N \right) \cdot U_h \, d\Omega + \int_{\Gamma_h} \Psi_h \cdot y \, (\dot{x}_N \, n)_{\text{down}} \, f_1(U_0) \, d\sigma \\
& + \int_{\Sigma_h} \left( \Psi_h^+ - \Psi_h^- \right) \cdot y \left( f_1^+(U_h) \, (\dot{x}_N \, n^+)_{\text{up}} + f_1^-(U_h) \, (\dot{x}_N \, n^-)_{\text{up}} \right) d\sigma \\
& + \int_\Omega \frac{\partial \Psi_h}{\partial x} \cdot f_2(U_h) \, d\Omega - \int_{\Gamma_h} \Psi_h \cdot f_2(U_0) \, n \, d\sigma \\
& - \int_{\Sigma_h} \left( \Psi_h^+ - \Psi_h^- \right) \cdot \frac{1}{2} \left( f_2(U_h)^+ + f_2(U_h)^- \right) n^+ \, d\sigma \\
& + \int_\Omega \Psi_h \cdot x_N f_3(U_h) \, d\Omega \,,
\end{aligned}
\tag{37}
$$

where $(\cdot)^+$, and $(\cdot)^-$ indicate the function values on either side of an interior element boundary.
$(\cdot)_{\text{up}}$ is equal to $(\cdot)$ where $\dot{x}_N \, n^\pm > 0$ and zero otherwise. Conversely, $(\cdot)_{\text{down}}$ is equal to $(\cdot)$ where $\dot{x}_N \, n^\pm < 0$ and zero otherwise.

$K(U)$ can be described in weak discretised form as

$$
\int_\Omega \Psi_h \cdot K(U_h) \, d\Omega = \int_\Omega \Psi_h \cdot K_\Omega(U_h) \, d\Omega + \int_{\partial\Omega_R} \Psi_h \cdot K_\sigma(U_h) \, d\sigma \,,
\tag{38}
$$

$$
K_\Omega(U_h) = \left[ 0, \, 0, \, 0, \, 0, \, 0, \, 0, \, C \, \frac{x_N \Delta_y}{\dot{x}_N} \right] \,,
\tag{39}
$$

$$
K_\sigma(U_h) = \left[ 0, \, 0, \, 0, \, 0, \, 0, \, Fr(\varphi)^{1/2}, \, 0 \right] \,,
\tag{40}
$$

where $\partial\Omega_R$ is the right-hand boundary at $y = 1$ such that a solution for $\dot{x}_N$ is obtained by solving only at the front of the current.

Using (30), (37) and (38), and applying (35) the full weak, discontinuous form of (24) can be obtained:



Find $U_h^{(0)}, U_h^{(1)}, U_h^{(2)}, U_h^{(n+1)} \in \mathcal{V}_h$ such that $\forall\, \Psi_h \in \mathcal{V}_h$

$$\int_\Omega \Psi_h \cdot U_h^{(0)}\, \mathrm{d}\Omega = \int_\Omega \Psi_h \cdot U_h^n\, \mathrm{d}\Omega\,,$$

$$\int_\Omega \Psi_h \cdot U_h^{(1)}\, \mathrm{d}\Omega = \int_\Omega \Psi_h \cdot A(U_h^{(0)})\, \mathrm{d}\Omega +$$

$$\int_\Omega \Psi_h \cdot L(U_h^{(0)})\, \mathrm{d}\Omega + \int_\Omega \Psi_h \cdot K(U_h^{(1)})\, \mathrm{d}\Omega\,,$$

$$\int_\Omega \Psi_h \cdot U_h^{(2)}\, \mathrm{d}\Omega = \int_\Omega \Psi_h \cdot A(U_h^{(1)})\, \mathrm{d}\Omega + \qquad (41)$$

$$\int_\Omega \Psi_h \cdot L(U_h^{(1)})\, \mathrm{d}\Omega + \int_\Omega \Psi_h \cdot K(U_h^{(2)})\, \mathrm{d}\Omega\,,$$

$$\int_\Omega \Psi_h \cdot U_h^{n+1}\, \mathrm{d}\Omega = \frac{1}{2}\int_\Omega \Psi_h \cdot \left(U_h^{(0)} + U_h^{(2)}\right)\, \mathrm{d}\Omega\,.$$

This set of equations is solved for each timestep of the simulation as a non-linear variational problem using Newton's method with an LU decomposition solver for the linear problems.

### 2.3  Slope limiting

Discontinuous Galerkin discretisations for convection dominated problems can suffer from over- and under-shoots at discontinuities that can cause instability problems (Kuzmin, 2010; Cockburn and Shu, 2001). Slope limiting can be applied to solve this problem, but this typically involves discontinuous operations, which are problematic in a gradient-based optimisation framework. Therefore, we do not use slope limiting here and limit ourselves to the assumption of smooth initial conditions

where slope limiting is not necessary. It would be possible to formulate a continuous slope-limiting function to overcome this limitation if it was required.

### 2.4  Implementation

The shallow water sediment-laden density current model described above was built using the FEniCS framework (Logg et al., 2012), an open-source software project that provides features for the

automated, efficient solution of differential equations. Using a high-level interface, the model partial differential equations are described in variational form using UFL (Unified form language) (Alnæs et al., 2012). This can be achieved in Python or C++ code in a way that is remarkably similar to how one would describe the equations on paper. At run-time this model description is compiled into efficient C++ kernels that handle assembly of the required matrices to generate the systems of equations

that are then solved using PETSc (Balay et al., 2014).





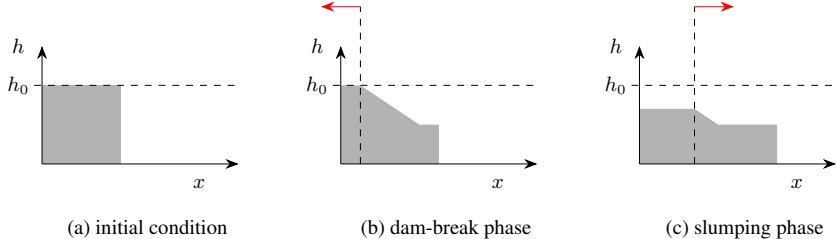

|                        |                     |                    |
| :--------------------: | :-----------------: | :----------------: |
| (a) initial condition  | (b) dam-break phase | (c) slumping phase |

**Figure 2.** Schematic diagram of the lock-release static initial condition (a) and the following dam-break (b) and slumping (c) phases with shock wave propagation direction indicted ( $\longrightarrow$ ).

## 2.5 Forward model verification

Many laboratory experiments and computer models are based around the classical lock-release static initial condition (Figure 2a). Following release of the lock-gate the current accelerates forwards. This is known as the dam-break stage (Ungarish, 2010) (Figure 2b). As the lock-gate is released a shock forms which travels in the opposite direction to the front of the current. This shock carries information that sets the fluid in motion. Once this shock reaches the rear wall, all of the fluid behind the lock-gate is in motion. This marks the point of transfer from the dam-break to the slumping phase (Ungarish, 2010) (Figure 2c). For a non-depositional current (i.e. $\beta = 0$) with initial $h$ and $x_N = 1$, the slumping phase begins at $t = 1$. The current front height and velocity remain approximately constant during this phase of motion. The rear propagating shock is reflected off of the no flow boundary and travels faster than the front of the current. A short while later it reaches the front of the current marking the end of the slumping phase. The current is now able to 'forget' the initial condition and begins adjusting to self-similar propagation (Ungarish, 2010). For a non-depositional current (i.e. $\beta = 0$) the reflected shock reaches the front of the current at $t = 3$ (Ungarish, 2010).

Hoult (1972) showed that a similarity solution could be obtained for a single-layer shallow water density current model during the self-similar phase of propagation. This is described as

$$x_N = \kappa t^{2/3}, \ \ u = \frac{2}{3}\kappa t^{-1/3}u_s, \ \ h = \kappa^{-1} t^{-2/3} h_s, \ \ \psi = 1 \,, \tag{42}$$

where

$$
\begin{aligned}
y &= x/\kappa t^{2/3}, & \kappa &= \left( \frac{27 Fr^2}{12 - 2Fr^2} \right)^{1/3}, \\
u_s &= y, & h_s &= \frac{4}{9}\kappa^3 \left( \frac{y^2}{4} - \frac{1}{4} + \frac{1}{Fr^2} \right).
\end{aligned}
\tag{43}
$$

The domain is unit length, as in all cases for this model. This solution is valid for the model described in this paper so long as the settling velocity of particles, $\beta$, is equal to zero (i.e. no particle settling). This analytical solution is useful in verifying the implementation of the governing equations and boundary conditions for this model. The solution to the model PDEs should converge to



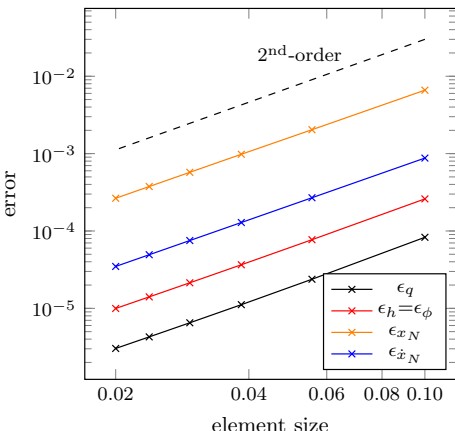

**Figure 3.** Similarity convergence analysis. All variables are shown to converge on the correct solution at the correct order. $\epsilon_{(*)}$ indicates the L$_2$-norm of the error in the solution obtained for variable $(*)$.

this analytical solution as the mesh resolution is refined, at the correct rate for the temporal, and
spatial discretisation. The use of piecewise discontinuous linear elements and a second-order time
stepping regime mean that the convergence order should be quadratic in both space and time.

For the convergence test the analytical solution is projected on to the model function space forming
the initial condition at $t = 3$. At $t = 10$ the $L_2$ norm of the difference between the model variables
and the analytical solution is obtained and used to measure convergence. The analysis shows that all
variables converge on to the analytical solution at the correct order (Figure 3). Note that the time step
is adaptive and will therefore decrease along with the element size such that this test is checking both
spatial and temporal convergence. This verifies that the model equations are implemented correctly
(Farrell et al., 2011). Qualitative comparison shows that the solution matches the analytical solution
very well (Figure 4).

## 3 The adjoint model

Here we describe the adjoint model and its derivation generally rather than specifically applying it
to this model.

Consider a problem with $N$ input parameters, forming a vector, $m$. Let $F(U, m) = 0$ denote the
set of PDEs that describe a model where $U$ represents the model variables throughout time. Note that
$U$ can be seen as an implicit function of $m$, $U = U(m)$, through finding a solution to $F(U, m) = 0$.
Suppose now that the aim is to minimise an objective functional, $J(U, m)$ by optimising $m$. Here,
$J(U, m)$, will be defined as a function measuring the difference between a deposit profile generated
by the model and a target deposit profile. Where optimisation is required to a tolerance of $\delta_{mi}$,





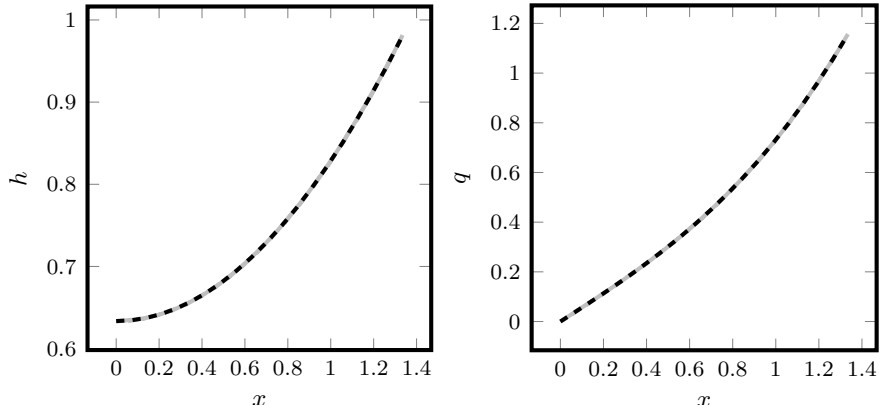

**Figure 4.** Similarity results for the finest resolution mesh (solid lines), compared against the analytical results (dashed lines) at $t = 10.0$.

with $i$ being the index of each parameter, and where each parameter has bounds spanning a range
$\Delta_{mi}$, optimisation by a brute force approach will require $\prod_i^N \Delta_{mi}/\delta_{mi}$ evaluations of the model to find the solution. $N$ may be very large for a sediment-laden density current model which could potentially have time-varying boundary conditions for sediment concentration, velocity and height, uncertainty in the elevation profile and friction coefficient of the surface over which the current is flowing, and uncertainty in parameters that govern physics of the flow such as entrainment of ambient
fluid, front speed, and sediment erosion. Many of these parameters vary over space and time such that the parameter space grows as the resolution in time or space increases. Such a large potential parameter space motivates the use of a more advanced and efficient optimisation strategy.

Numerous algorithms have been developed that improve on this brute force approach. These optimisation algorithms begin with an initial guess of the input parameters and iterate, generating
improved estimates until they terminate, hopefully at the optimised solution. The authors refer the reader to Jorge and Stephen (1999) for an extensive description of the range of numerical optimisation methods.

Most of these optimisation algorithms require the gradient of the objective functional with respect to the input parameters, $\mathrm{d}J/\mathrm{d}m$. Approximation techniques, such as finite-differencing, could be
used to evaluate the gradient, but this will require an excessive number of PDE evaluations and may suffer from noise (Jorge and Stephen, 1999). Here, the adjoint model is used to efficiently calculate the gradient. This approach is favoured as it calculates $\mathrm{d}J/\mathrm{d}m$ for any number of input parameters with a single evaluation of the adjoint model.





Obtaining the adjoint model begins by applying the chain rule to $\mathrm{d}J/\mathrm{d}m$

$$\frac{\mathrm{d}J(U(m),m)}{\mathrm{d}m} = \left\langle \frac{\partial J}{\partial U}, \frac{\mathrm{d}U}{\mathrm{d}m} \right\rangle + \frac{\partial J}{\partial m} \, . \tag{44}$$

$\partial J/\partial U$ and $\partial J/\partial m$ are both vectors, and they are typically straightforward to compute as $J$ is typically a given analytical function of $U$ and $m$. $\mathrm{d}U/\mathrm{d}m$ on the other hand is a matrix that is typically dense and is expensive to compute. A relationship for $\mathrm{d}U/\mathrm{d}m$ can be obtained by taking the total derivative of $F(U,m) = 0$ with respect to $m$

$$0 = \frac{\mathrm{d}F(U(m),m)}{\mathrm{d}m} = \frac{\partial F}{\partial U}\frac{\mathrm{d}U}{\mathrm{d}m} + \frac{\partial F}{\partial m} \, , \tag{45}$$

$$\Rightarrow \frac{\partial F}{\partial U}\frac{\mathrm{d}U}{\mathrm{d}m} = -\frac{\partial F}{\partial m} \, . \tag{46}$$

Equation 46 is termed the *tangent-linear equation*. $\partial F/\partial U$ and $\partial F/\partial m$ are both matrices. The solution of this equation is obtained by solving $N$ systems of equations. When there are many functionals, $J$, and a small set of parameters, $m$, then this equation can be useful for obtaining $\mathrm{d}J/\mathrm{d}m$

via (44). With a large set of parameters and only one functional, as is the case here, this is not an efficient approach.

However, suppose that $\partial F/\partial U$ in (46) is invertible so that one can obtain

$$\frac{\mathrm{d}U}{\mathrm{d}m} = -\left(\frac{\partial F}{\partial U}\right)^{-1}\frac{\partial F}{\partial m} \, . \tag{47}$$

This expression can be substituted for $\mathrm{d}U/\mathrm{d}m$ directly into (44) to obtain

$$\frac{\mathrm{d}J(m)}{\mathrm{d}m} = -\left\langle \frac{\partial J}{\partial U}, \left(\frac{\partial F}{\partial U}\right)^{-1}\frac{\partial F}{\partial m} \right\rangle + \frac{\partial J}{\partial m} \, . \tag{48}$$

A simple property of inner products, $\langle y, Ax\rangle = \langle A^*y, x\rangle$, where $A^*$ is the conjugate transpose, or adjoint, of $A$, can be used to shift $(\partial F/\partial U)^{-1}$ to the left hand side of the inner product

$$\frac{\mathrm{d}J(m)}{\mathrm{d}m} = -\left\langle \left(\frac{\partial F}{\partial U}\right)^{-*}\frac{\partial J}{\partial U}, \frac{\partial F}{\partial m} \right\rangle + \frac{\partial J}{\partial m} \, . \tag{49}$$

Gathering the left-hand side of the inner product into a new variable,

$$\lambda := \left(\frac{\partial F}{\partial U}\right)^{-*}\frac{\partial J}{\partial U} \, , \tag{50}$$

yields the linear system of equations that can be solved for the adjoint variable, $\lambda$

$$\left(\frac{\partial F}{\partial U}\right)^{*}\lambda = \frac{\partial J}{\partial U} \, . \tag{51}$$

Equation (51) is termed the *adjoint equation*. The right hand side is a vector and only one evaluation is required to obtain $\lambda$ for a specific functional, $J$. Once (51) is solved, $\mathrm{d}J/\mathrm{d}m$ can easily be

computed with respect to any parameter $m$ by substituting the value of $\lambda$ into (48).





As commented above, $\partial J/\partial U$ and $\partial J/\partial m$ are typically straightforward to compute. However, $(\partial F/\partial U)^*$ and $\partial F/\partial m$ still need to be derived and implemented which is not a simple task for a large set of complex PDEs. The challenge of obtaining these matrices is the main obstacle to using the adjoint model. However, the high-level abstraction of the coding provided by using FEniCS to create this model makes calculating $(\partial F/\partial u)^*$ and $\partial F/\partial m$ an automatable task using an additional tool, dolfin-adjoint (Farrell et al., 2013). This powerful tool automatically derives the discrete adjoint and tangent linear models from a forward model written in FEniCS. This makes differentiating the forward model, and solving the adjoint equation to obtain the derivative of the objective functional a much simpler task. Additionally to this, dolfin-adjoint also contains tools for carrying out opti-misation of model parameters by interfacing with IPOPT (Wächter and Biegler, 2006) optimisation algorithms (Funke and Farrell, 2013).

## 4 Estimation of parameters for the turbidity current that generated Bed 1.1 in the Marnoso Arenacea Formation

The Marnoso Arenacea Formation spans 17 to 7 Ma (Late Burdigalian to Tortonian) and is over 3500m thick (Talling et al., 2007b). Deposition occurred from two sources: the northwestern Alpine source and the southwestern Apennine source (Lucchi and Valmori, 1980; Gandolfi et al., 1983). The depositional environment was an elongate foreland basin adjacent to the Apennine thrust belt with turbidites being deposited in a relatively wide (>60km) basin, in a non-channelised manner (Talling et al., 2007b; Lucchi and Valmori, 1980; Gandolfi et al., 1983). The formation provides the most extensive and detailed correlation of flow deposits (beds) in any ancient turbidite system and is therefore a natural laboratory for studying turbidite depositional processes (Amy and Talling, 2006). It has been extensively mapped with more than 100 sections being accurately recorded over a corre-lated distance of more than 120 km (Amy and Talling, 2006). Bed volumes range from $\mathcal{O}(10^{-3})\text{km}^3$ to several $\text{km}^3$ (Talling et al., 2007a). It contains extensive data for evaluating the performance of the adjointed turbidity current model described here.

In this section an optimisation algorithm is used to select model parameters that produce an output deposit that best matches part of Bed 1.1 in the Marnoso Arenacea Formation, as recorded by Amy and Talling (2006). This is defined as a small volume flow deposit with a total sediment volume of $\approx 0.215\text{km}^3$ (Talling et al., 2007a). Talling et al. (2007a) produced an approximate one-dimensional deposit parallel to the palaeoflow (Figure 5). The shape of the deposit strongly resembles that of very low concentration currents in laboratory tests, and also resembles the shape of bed profiles generated by the Bonnecaze et al. (1993) model. This implies that the flow that created this deposit was a very low concentration current. The model used in this chapter is very simple. It does not model any stratification, or particle-particle interactions in the flow. As such its application is limited to very low concentration flows and hence Bed 1.1 is a good candidate case study for this model.




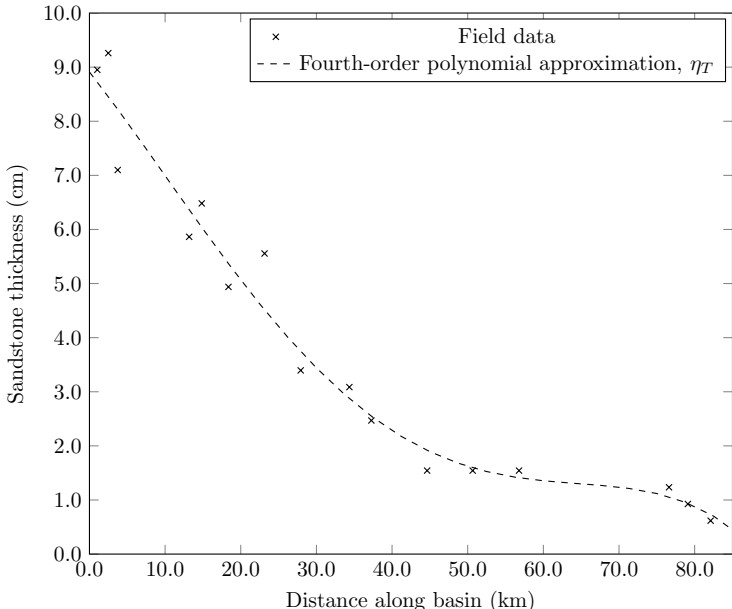

**Figure 5.** Sandstone depths measured for Bed 1.1 along the Pietralunga and Ridracoli structural elements orientated approximately parallel to the palaeoflow. This has been reconstructed from Figure 5 in Talling et al. (2007a). A fourth-order polynomial approximation of the deposit profile, $\eta^T$ is also shown. This is used as a target for the optimisation algorithm. The base of the bed is shown as a horizontal datum in order to illustrate lateral changes in deposit thickness. Note that a different datum is used in the source figure, which uses the top of Bed 1.2.2 rather than the top of Bed 1. The palaeo-elevation of the base of the bed would have varied spatially, reflecting basin-floor relief.

The deposit consists of sandstone and mudstone components. The focus here will be on attempting to recreate only the sandstone portion of the deposit. It is likely that ponding effects have influenced the shape of the mud deposit in this bed (Talling et al., 2007b) which this model cannot replicate. The outcrop quality also deteriorates beyond the extent of the sandstone deposit. Therefore, no attempt 400 is made to model this portion of the bed.

### 4.1 Choice of initial conditions and parameters

The initial conditions are based upon the analytical solution for a non-depositional flow at a non-dimensional time, $t = t_s = 3$ after a column collapse, as described by (42). Some assumptions are therefore made as to the initial shape, sediment concentration profile, and velocity profile of the flow.





The non-dimensional particle settling velocity, $\beta$ is calculated using the standard Stokes settling
law for a particle in suspension (Lamb, 1993), non-dimensionalised by $(h_0\, g_0')^{1/2}$ to give

$$\beta = \frac{g'\, D^2}{18\,\nu\,(h_0\, g_0')^{1/2}} = \frac{g'^{1/2}\, D^2}{18\,\nu\,(h_0\,\psi_0)^{1/2}} \,, \tag{52}$$

where $D$ is the average sediment diameter. The sediment reduced gravity, $g' = g_0'/\psi_0 = (\rho_p - \rho_a)\, g/\rho_a =$
16, is based upon the reduced gravity of silica in water. Using these initial conditions there are three
unknowns, $h_0$, the dimensional length scale of the current, $\psi_0$, the initial sediment concentration
throughout the current, and $D$, the mean sediment diameter. These become the set of input parame-
ters that will be optimised, $m = (h_0, \psi_0, D)^T$.

The beginning of the basin is defined as being at the front of the current at $t = t_s$ such that the
current is not in the basin prior to the start of the simulation. The current enters the domain as soon as
the simulation starts. The end of the simulation, $t_f$, is defined as the time at which the total suspended
sediment is less than $1\%$ of the starting quantity.

### 4.2 Choice of optimisation functional

The aim here is to reduce the difference between the deposit profile generated by this model and
the target deposit profile from field measurements. To do this we need to map the non-dimensional,
transformed results from the model back to the observation space. We also only measure the variation
over the length of the measured deposit. Therefore, the functional that we will aim to minimise, $J$,
has the form

$$J(U(m), m) = \int\limits_{0}^{\hat{x}_{\max}} (\tilde{\eta} - \eta_T)^2 \, \mathrm{d}\hat{x} \,, \tag{53}$$

where $\eta_T$ is the dimensional target deposit profile, $\tilde{\eta} = \psi_0\, h_0\, \eta$, is the dimensionalised modelled
deposit, $\hat{x}_{\max} = 82000$, is the extent of the measured data, and $\hat{x} = \tilde{x} - \tilde{x}_N(t_s)$, is a coordinate
transformation such that $\hat{x} = 0$ at the front of the current at $t = t_s$, $\tilde{x} = y\, \tilde{x}_N(t_f)$ is the dimension-
alised reverse of the coordinate transformation outlined in Section 2.1, and $\tilde{x}_N(t) = h_0\, x_N(t)$, is the
dimensional length of the current.

To calculate this functional, $\eta^T$ must be a function of $\hat{x}$. The deposit is approximated using a
fourth-order polynomial as

$$\eta^T = \sum_{i=0}^{4} c_i\, \hat{x}^i \,, \tag{54}$$

where $c_i$ is the $i^{\text{th}}$ coefficient. The coefficients are obtained using the least squares method. The
fourth-order approximation fits the measured data points well (Figure 5).

It is important to note that at the end of the simulation, $t = t_f$, the length of the current does not
necessarily match the length of the deposit, or $\hat{x}_N \neq \hat{x}_{\max}$, where $\hat{x}_N(t) = \tilde{x}_N(t) - \tilde{x}_N(t_s)$, is the





dimensional length of the modelled deposit within the basin. This complicates the calculation of the above integral.

Calculation of $J$ is split in to two components, an integral over the lesser of the length of the modelled current, or the length of the measured data, $J_0$, and an integral over any remaining length of measured data, $J_1$, such that

$$J = J_0 + J_1 \,. \tag{55}$$

The first integral takes the form

$$J_0 = \int_0^{\min(\hat{x}_N, \hat{x}_{\max})} (\eta_T(\hat{x}) - \tilde{\eta}(\hat{x}))^2 \, \mathrm{d}\hat{x} \,, \tag{56}$$

This can be approximately transformed in to the model coordinate system as

$$J_0 = \int_\Omega \left( \gamma_0 \, \eta_T(\hat{x}(y)) - \gamma_0 \, \tilde{\eta}(\hat{x}(y)) \right)^2 \, \mathrm{d}y \tag{57}$$

where $\gamma_0$ is a scaled filter. This filter is zero in the region $\hat{x} < 0$ and $\hat{x} > \hat{x}_{\max}$. Elsewhere the filter value is a constant such that the integral of the filter over the domain is equal to the length of the dimensional integral, $\min(\hat{x}_N, \hat{x}_{\max})$. As such the filter defines the region of the domain over which the integral is evaluated and scales the resultant value appropriately. The filter is defined as

$$\gamma_0(\hat{x}) = \max(\hat{x}_N, \hat{x}_{\max}) \frac{\exp\left( \min(\hat{x} - \hat{x}_{\max}, \hat{x}, 0) \right)}{s_{\gamma_0}} \,, \tag{58}$$

$$s_{\gamma_0} = \int_\Omega \exp\left( \min(\hat{x} - \hat{x}_{\max}, \hat{x}, 0) \right) \, \mathrm{d}\hat{x} \,. \tag{59}$$

It is important the functional is differentiable. Therefore the $\min$ and $\max$ functions are replaced by smooth approximations $f_{\min}$ and $f_{\max}$ defined as

$$f_{\min}(a, b) = \ln(\exp(10\,a) + \exp(10\,b))/10 \,, \tag{60}$$

$$f_{\max}(a, b) = f_{\min}(-a, -b) \,. \tag{61}$$

The second integral, $J_1$, takes the form

$$J_1 = \int_{\min(\hat{x}_N, \hat{x}_{\max})}^{\hat{x}_{\max}} \eta_T(\hat{x})^2 \, \mathrm{d}\hat{x} \,, \tag{62}$$

such that $J_1$ integrates the target deposit volume beyond the extent of the modelled deposit. If the modelled deposit length exceeds the length of the measured data this integral will be zero. Again, this can be approximately transformed in to the model coordinate system as

$$J_1 = \int_\Omega \left( \gamma_1 \, \eta_T(\tilde{x}(y)) \right)^2 \, \mathrm{d}y \,, \tag{63}$$

$$\tilde{x}(y) = \min(\hat{x}_N + y(\hat{x}_{\max} - \hat{x}_N), \hat{x}_N) \,, \tag{64}$$



where $\gamma_1$ is a scaled filter similar to $\gamma_0$ defined as

$$\gamma_1(\hat{x}) = (\hat{x}_{\max} - \min(\hat{x}_N, \hat{x}_{\max})) \, \frac{\exp\left(\,\min(\hat{x} - \hat{x}_{\max}, 0)\,\right)}{s_{\gamma_0}} \,, \tag{65}$$

$$s_{\gamma_0} = \int_\Omega \exp\left(\,\min(\hat{x} - \hat{x}_{\max}, 0)\,\right) \, \mathrm{d}\hat{x} \,, \tag{66}$$

Again, $\min$ and $\max$ are replaced by smooth differentiable alternatives.

### 4.3 Verification of the gradient calculation

The gradient computation was verified using the Taylor remainder convergence test. Let $\widehat{J}(m) \equiv J(U(m), m)$, a pure function of $m$. The first-order Taylor expansion states that:

$$\left| \widehat{J}(m + \delta m) - \widehat{J}(m) - \frac{\mathrm{d}\widehat{J}(m)}{\mathrm{d}m} \delta m \right| = \mathcal{O}\left( \|\delta m\|^2 \right) \,. \tag{67}$$

The above calculation was carried out for a number of perturbation directions and magnitudes about a number of choices for $m$. The convergence of the remainder term was second order with respect to varying magnitudes of $\delta m$, for all perturbation directions, providing strong evidence that the adjoint model and gradient computation are implemented correctly.

### 4.4 Optimisation of a model with one sediment class

With confidence that the forward and backward models are working, optimisation of the input parameters, $m = [h_0, \psi_0, D]$, to minimise the objective functional $J$ can now be performed.

$$\min_m = J(U(m), m) \,, \tag{68}$$

with the the following bounding constraints on the input parameters

$$10 \, \mathrm{m} \leq h_0 \leq 10 \, \mathrm{km} \,, \tag{69}$$

$$0.001\% \leq \psi_0 \leq 50\% \,, \tag{70}$$

$$1 \, \mu\mathrm{m} \leq D \leq 1 \, \mathrm{mm} \,. \tag{71}$$

These bounding constraints are chosen based upon very loose limits of expected values that each parameter may possibly take. The principal purpose of these bounds is to avoid invalid negative
values being generated for any of the parameters.

The nonlinear optimisation library, IPOPT (Wächter and Biegler, 2006), is used to solve this problem. This library implements a primal-dual interior-point algorithm which has good global and local convergence properties (Wächter and Biegler, 2005). The interface to this library is supplied by dolfin-adjoint (Funke and Farrell, 2013).





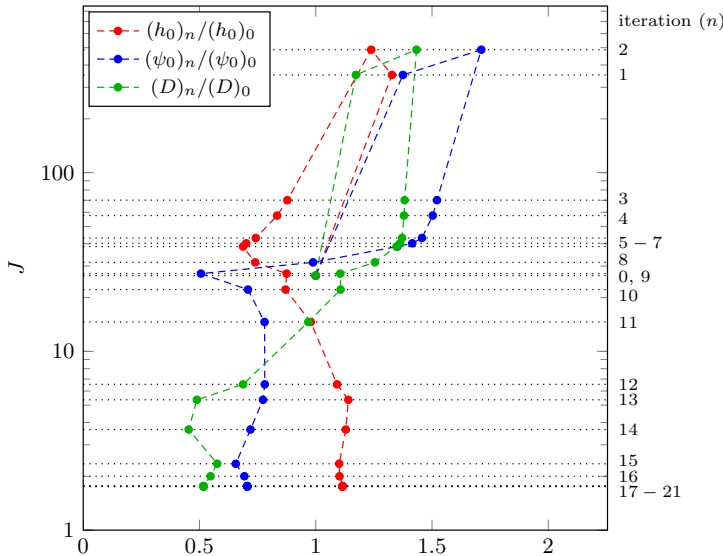

**Figure 6.** Values of parameters over the optimisation iterations against the value of the objective functional, $J$, that we are aiming to minimise. Values shown are normalised by their starting values. $(*)_n$ is the value of parameter $*$ at the start of iteration $n$.

The initial input parameters are set to

$$m = \begin{pmatrix} h_0 & = & 2.3\,\text{km} \\ \psi_0 & = & 0.07\% \\ D & = & 200.0\,\mu\text{m} \end{pmatrix}. \tag{72}$$

The aim is to recreate the sand deposit by modelling only the sand in the flow using a single average grain size, $D$. The value of $\psi_0$ is based upon a combined initial volumetric concentration for the sand and mud mixture of $0.5\%$, with $86\%$ of the mixture being mud. The starting value for $h_0$ is based upon the area of the two-dimensional deposit profile and the value of $\psi_0$. The average sediment grain size is a reasonable estimate of the average grain size based upon the information provided by Talling et al. (2007a). The input parameters provided to the optimisation algorithm, $\bar{m}$, are normalised such that they are all equal to one. Thus

$$m_i = \bar{m}_i \, m_0 \tag{73}$$

where $m_0$ are the initial parameter values in (72), and $m_i$ indicates the value of $m$ after optimisation iteration $i$. This scaling helps the optimisation algorithm work effectively (Jorge and Stephen, 1999).

The criteria for finishing the parameter optimisation is based upon the relative change in $J$ between iterations such that

$$\frac{|J_i - J_{i-1}|}{J_i} < 1.0 \times 10^{-5}, \tag{74}$$



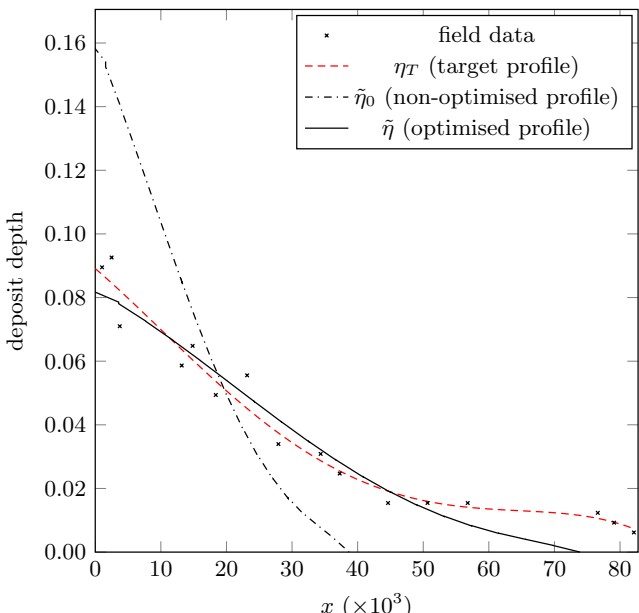

**Figure 7.** Dimensional deposit output from initial parameter guess, $\tilde{\eta}_0$, and optimised dimensional deposit output, $\tilde{\eta}$, shown against the field measurement from Bed 1.1 (Talling et al., 2007a) and fourth-order polynomial target deposit profile, $\eta^T$.

where $J_i$ is the value of $J$ after the $i^{th}$ iteration. The optimisation is completed in 21 iterations, with a final functional value of $J = 1.75$ (Figure 6). The optimised deposit profile, $\eta$, compares relatively well with $\eta^T$ (Figure 7). Most notably there is a significant variation in the thickness towards the end of the deposit. This will be addressed later.

The final optimised values are

$$m = \begin{pmatrix} h_0 & = & 2.56\,\mathrm{km} \\ \psi_0 & = & 0.0494\% \\ D & = & 103\,\mu\mathrm{m} \end{pmatrix} . \qquad (75)$$

These optimised values are not completely acceptable. The value for $h_0$ represents the initial height of the current if it started from a static lock-release initial condition. This translates to an initial current height of 993.3m at the start of this simulation, and as the current enters the basin plain. This value appears to be quite large for a relatively small turbidity current. Additionally, the average sediment diameter of $103\mu$m is lower than expected. Talling et al. (2007a) defines the sandstone interval as being dominated by sediment grains estimated to be larger than $\approx 125\mu$m.

With the exception of the sediment diameter, the optimised values are fairly similar to those chosen as input values. This confirms that the input parameters chosen were sensible predictions of the





starting conditions for the gravity current. It is possible that the optimisation algorithm has obtained

a local minimum rather than a global minimum and that different starting values would produce a

different result.

A clear omission from the model is the presence of mud in the suspension. The presence of mud will significantly alter the energy budget of the flow. A mud sediment class can easily be included such that the model produces more realistic optimised values. This is detailed below.

**4.5 Extending the model to include an additional sediment class for the mud in suspension**

Investigating the effect of including mud in the sediment mixture can be achieved relatively simply by including an additional transport equation with a form identical to (12) (Dorrell et al., 2013)

$$\frac{\partial \varphi_m}{\partial t} = \frac{1}{x_N} \left( y \dot{x}_N \frac{\partial \varphi_m}{\partial y} - \frac{\partial}{\partial y} \left( \frac{q \varphi_m}{h} \right) \right) - \beta_m \frac{\varphi_m}{h} \ , \tag{76}$$

where $\varphi_m = \psi_m h$ is the vertically-integrated volume fraction of mud in the suspension with $\psi_m$

being the depth-averaged volume fraction of mud within the flow, and $\beta_m$ is the settling velocity of the mud particles. Using a single tracer equation we approximate the distribution of mud particle sizes using a single mud diameter, in the same way as the distribution of sand is modelled in the flow. We neglect flocullation of mud particles. Assuming that the density of both sediment classes are the same, (11) is modified to include this new sediment class in the gravity term

$$\frac{\partial q}{\partial t} = \frac{1}{x_N} \left( y \dot{x}_N \frac{\partial q}{\partial y} - \frac{\partial}{\partial y} \left( \frac{q^2}{h} + \frac{(\varphi + \varphi_m)h}{2} \right) \right) \ . \tag{77}$$

Finally $\varphi$ and $\varphi_m$ are scaled such that at the start of the simulation $\varphi + \varphi_m = 1$, where previously $\varphi = 1$. The aim is still to recreate the deposit of sand and hence the equation for $\eta$ stays the same. We term the sand deposit generated by this modified model $\eta_2$. The discretisation for (76) is consistent with the rest of the model, as presented in section 2.2.

The initial condition needs to be altered to include the new sediment class. The initial vertically-averaged volume fraction of sand is changed to be $\psi = f_s$, and a new initial condition for the vertically-averaged volume fraction of mud is introduced $\psi_m = 1 - f_s$. The sand fraction, $f_s$, is estimated by Talling et al. (2007a) to be $0.14$ and is kept fixed.

$\beta_m$ must also be calculated. This is done in the same way as for $\beta$ except that a different sediment

diameter parameter, $D_m$, the mean diameter of mud particles in the flow, is used and optimised. The equation for $\beta_m$ is therefore

$$\beta_m = \frac{g'^{1/2} D_m^2}{18 \, \nu \, (h_0 \, \psi_0)^{1/2}} \ . \tag{78}$$

Note that $\psi_0$ is now the combined initial volume fraction of sand and mud in the flow.





### 4.6 Optimisation for a model with two sediment classes

The set of optimised input parameters is redefined as $m = [h_0, \psi_0, D, D_m]^T$. An additional bounding constraint is added for $D_m$ such that the new bounding constraints for $m$ are

$$10\,\mathrm{m} \le h_0 \le 10.0\,\mathrm{km}\,, \tag{79}$$

$$0.001\,\% \le \psi_0 \le 50\,\%\,, \tag{80}$$

$$1.0\,\mu\mathrm{m} \le D \le 1.0\,\mathrm{mm}\,, \tag{81}$$

$$1.0\,\mu\mathrm{m} \le D_m \le 100.0\,\mu\mathrm{m}\,. \tag{82}$$

The initial input parameters are set to

$$m = \begin{pmatrix} h_0 & = & 2.1\,\mathrm{km} \\ \psi_0 & = & 5\,\% \\ D & = & 200.0\,\mu\mathrm{m} \\ D_m & = & 20.0\,\mu\mathrm{m} \end{pmatrix}. \tag{83}$$

The input parameter are normalised as detailed in section 4.4 before being passed to the optimisation algorithm. The criteria for finishing the optimisation is consistent with the previous optimisation (see

560    (74)).

    Optimisation of the model with two sediment classes is completed in 17 iterations with a final functional value of $J = 2.13$ (see Figure 8). Therefore, quantitatively the fit is very slightly worse when mud is included in the model. This is a surprising result as the model now more closely matches reality. Qualitatively it is very hard to determine which model fits the data better. The resultant

deposit is very similar in shape to that obtained when only modelling sand in the flow (see Figure 9). The fit appears to be worse at the start of the deposit. The runout length is slightly longer when mud is included such that the fit towards the end of the deposit is slightly improved.

    The fit with the measured data is still poor towards the end of the deposit. Talling et al. (2007a) noted how the distal section of small deposits in the Marnoso Arenacea formation show evidence of

transport in a tractional boundary layer. This simulation does not model bedload transport or erosion which is the likely reason for the difference in results. The velocity of the head of the turbidity current in this simulation varies between $10\,\mathrm{ms}^{-1}$ and $2.4\,\mathrm{ms}^{-1}$ over the period where sand is deposited (Figure 11). At these head velocities erosion is very likely to occur. Models for erosion and bedload transport exist (Garcia, 1994; Sequeiros et al., 2009). These could be added in future work.

The final optimised values are

$$m = \begin{pmatrix} h_0 & = & 1920 \\ \psi_0 & = & 5.94 \times 10^{-3} \\ D & = & 125\,\mu\mathrm{m} \\ D_m & = & 28.1\,\mu\mathrm{m} \end{pmatrix}. \tag{84}$$





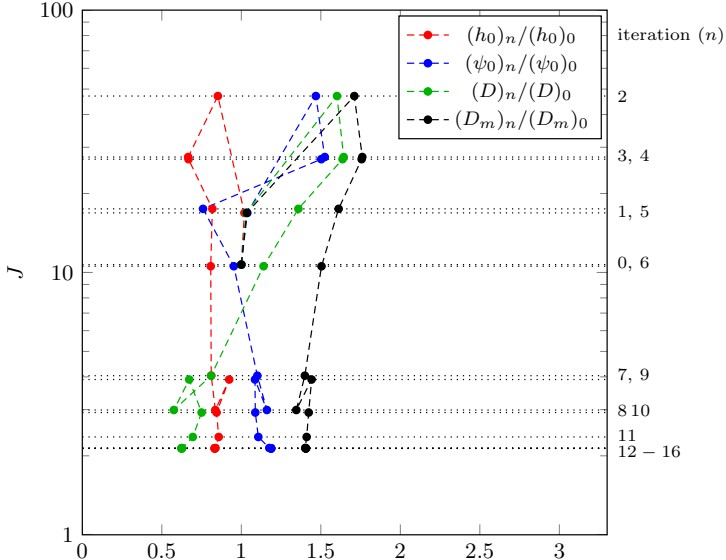

**Figure 8.** Values of parameters from the model with both mud and sand sediment classes over the optimisation iterations against the value of the objective functional, $J$, that we are aiming to minimise. Values shown are normalised by their starting values. $(*)_n$ is the value of parameter $*$ at the start of iteration $n$.

Comparing these results to those obtained without a mud sediment class, the value of $h_0$ has reduced by $25\%$ and translates to an initial current height of $745.0$m as the current enters the basin plain. The average sediment diameter has also increased by $21\%$ to $125\mu$m bringing the average diameter in line with the estimates from field measurements by Talling et al. (2007a). Arguably, the sand and mud classes should be subdivided further. Dorrell et al. (2013) described how polydisperse density currents will have longer run out distances than equivalent currents with uniform sediment at the mean value of the poydisperse current.

It is also interesting to assess the sensitivity of the model to variations in the input parameters by analysing the final gradient of the objective functional,

$$\frac{\mathrm{d}J}{\mathrm{d}\bar{m}} = \begin{pmatrix} \mathrm{d}J/\mathrm{d}\bar{h}_0 & = & 5.1 \times 10^{-3} \\ \mathrm{d}J/\mathrm{d}\bar{\psi}_0 & = & -1.8 \times 10^{-3} \\ \mathrm{d}J/\mathrm{d}\bar{D} & = & -2.4 \times 10^{-3} \\ \mathrm{d}J/\mathrm{d}\bar{D}_m & = & 1.5 \times 10^{-6} \end{pmatrix} . \tag{85}$$

where $\bar{\phantom{x}}$ indicates a parameter value normalised by its value on the initial optimisation iteration. The sensitivity of the functional to changes in the mud diameter is several orders of magnitude smaller than the sensitivity to changes in the other variables.



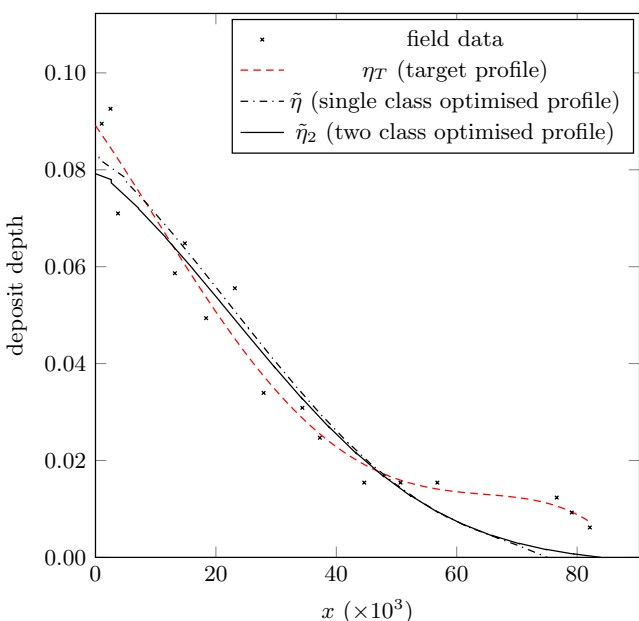

**Figure 9.** Optimised dimensional deposit output from the model with both mud and sand sediment classes, $\tilde{\eta}_2$ shown against the optimised results from the single sediment class model, $\tilde{\eta}$, the field measurement from Bed 1.1 (Talling et al., 2007a) and fourth-order polynomial target deposit profile, $\eta^T$.

It is indeed found that changing this value has very little effect on the obtained deposit. The same simulation is run with the mud diameter decreased by two orders of magnitude such that the input parameter values are

$$
m = \begin{pmatrix} h_0 &=& 1920 \\ \psi_0 &=& 5.94 \times 10^{-3} \\ D &=& 125\,\mu\mathrm{m} \\ D_m &=& 0.281\,\mu\mathrm{m} \end{pmatrix}. \tag{86}
$$

The resulting functional value, $J = 2.13$, which is identical to that obtained for the optimised sim-
ulation. There is no discernible difference in the resulting deposit, $\eta_3$ (Figure 10). The head height and velocity only vary a small amount over the period where sand is deposited (Figures 11a and 11b). The current properties vary significantly after the sand has been deposited and mud is still in suspension, but this does not have any effect on the sand deposit.

    Although the sandstone deposits generated by the single and two sediment class models are very
similar, properties of the turbidity currents that produced them are very different (Figure 11). The turbidity current with mud in suspension travels approximately twice as quickly due to the increased gravitational forces produced by the sand and mud mixture (Figure 11b). Sand also drops out of the



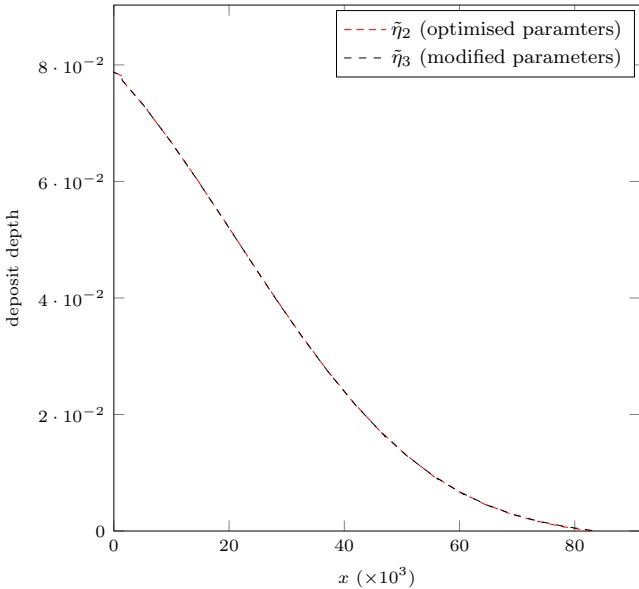

**Figure 10.** Optimised dimensional deposit output from the model with both mud and sand sediment classes, $\tilde{\eta}_2$ shown against results from the same model, with the same parameters but a mud settling velocity reduced by two orders of magnitude, $\tilde{\eta}_3$.

suspension much more rapidly (Figure 11d). All of the sand is deposited within approximately six hours. The model without mud deposits sand over a period of more than twenty hours. This is due to

the reduced height of the current at the start of the simulation, and the faster decrease in the height of the current as a result of the higher head velocity (Figures 11a and 11d). Clearly the presence of mud in the suspension has a significant impact on the resultant flow and must be included in the model.

The simulated turbidity currents that produced $\eta_2$ and $\eta_3$ deposited sand over a similar time period. After all of the sand had fallen out of suspension less than 25% of the mud has settled from the flow

for both of these currents, the current head is $> 50$m tall, and the head is moving at $> 1.0$m/s (Figure 11). Hence there is still a significant amount of energy in the flow. The remaining mud suspension will reach the end of the basin ($\tilde{x}_N \approx 130$km) and will still have a significant amount of energy left when it does so. It is very hard to predict what will happen after this point. The current may be partly reflected and ponding of the suspended mud is likely to occur. This result is in agreement with the

explanations of Talling et al. (2007b).

The height of the current in the optimised simulation with both sand and mud sediment classes is $\approx 750$m as it enters the basin, although this decreases very quickly as the current propagates. It is possible that including processes such as fluid entrainment, erosion, and bedload transport, may reduce the necessity for such a large initial current height in producing this deposit. More complex





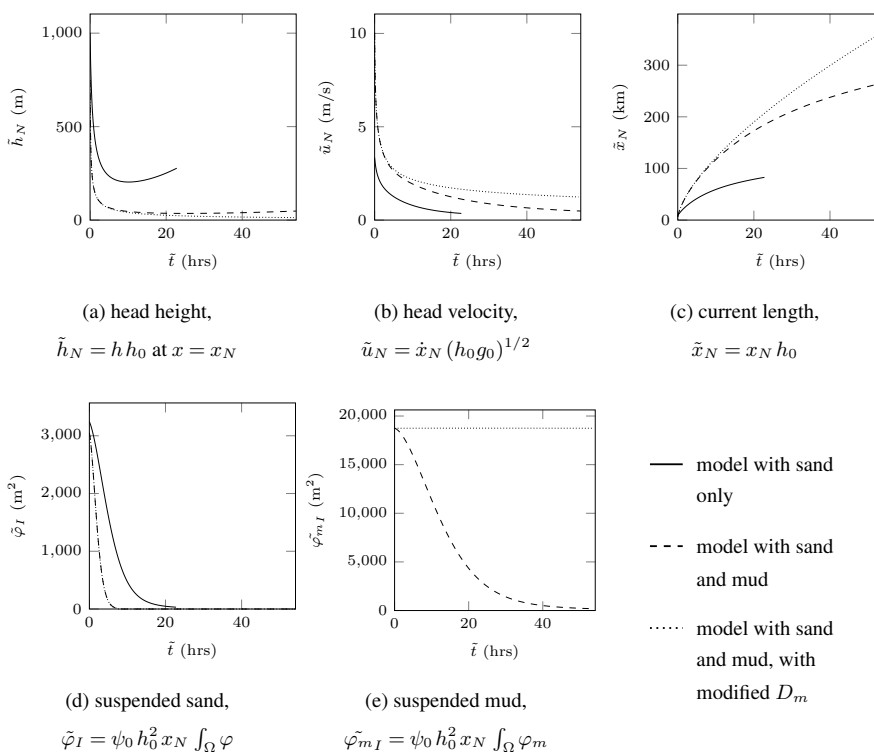

**Figure 11.** Time evolution of dimensionalised variables for three simulations: a simulation with a single sand sediment class and optimised input parameters to match the Bed 1.1 sand deposit, a simulation with sand and mud classes and optimised input parameters to match the Bed 1.1 sand deposit, and a simulation with sand and mud classes and the same optimised input parameters but a mud diameter two-orders of magnitude smaller. The results are shown against the dimensional time, $\tilde{t} = t \, (h_0/g_0)^{1/2}$.

initial and boundary conditions may also have a significant impact on this value. It is unclear what effect an inflow boundary condition with time-varying height, sediment concentration, and velocity would have on the results. This would be an interesting addition to the models capabilities.

The model also neglects variations in the bed profile. The gradient of the sea floor in the basin where the Marnoso Arenacea formation was created was substantially less than one degree (Amy

and Talling, 2006). Variations in gradient of this magnitude will have negligible impact on the head velocity (Middleton, 1966). However, small variations will have an impact on the velocity of the body of the current. Future work will address this.





## 5   Conclusions

This paper has presented a novel implementation of the shallow water equations for modelling density currents using a mixed finite element formulation. The model has been differentiated to allow for parameter optimisation using gradient-based optimisation techniques, and the use of gradient information in sensitivity analyses.

The proposed model is based upon simplifying shallow water sediment-laden density current assumptions, and has been used here to recreate a low volume deposit from the Marnoso Arenacea Formation, Italy with some success. However, the lack of many key flow processes within the current model, including bedload transport and reentrainment, has arguably led to optimised parameters values which would be improved upon with a more complete underlying model.

This paper has demonstrated the power of gradient-based optimisation methods for determining the set of input parameters that best fits a particular turbidity current deposit. Since input parameters are rarely known with any accuracy for these flows, optimisation represents a sensible way to better estimate these values.

Future development of the model could enable more complex boundary conditions and will include the addition of parameterisations for ambient fluid entrainment, bed erosion, and bedload transport. This will increase the capacity for the model to recreate a range of deposits found in the field, while the parameter space will grow significantly. The optimisation techniques presented in this paper will allow for efficient selection of optimised values for a large parameter space.

### Code availability

The model implementation and the test setups described in this paper are freely available as a separate git repository on bitbucket: https://bitbucket.org/simon_funke/adjoint-sw-sediment. This repository contains a README file which guides the user through the installation and how to reproduce the results of the paper. The experiments configuration files can viewed and changed with the graphical configuration tool spud from the Fluidity project (http://fluidityproject.github.io/). The dynamical core of the model is implemented with the finite element software FEniCS and its extension dolfin-adjoint. The documentation for FEniCS is available on http://fenicsproject.org/documentation. and the documentation for dolfin-adjoint can be found on http://www.dolfin-adjoint.org.

All FEniCS core components and dolfin-adjoint are licensed under the GNU LGPL as published by the Free Software Foundation, either version 3 of the license, or (at your option) any later version.

*Acknowledgements.* This work was supported by the Natural Environment Research Council grant NE/K000047/1 and the Research Council of Norway through a Centres of Excellence grant to the Center for Biomedical Computing at Simula Research Laboratory (project number 179578).



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
