# Peer review of "Application of the adjoint approach to optimise the initial conditions of a turbidity current"

_Geoscientific Model Development, 2016_

## Short Comment (SC1) · 20 Jul 2016

Dear authors,

In my role as Executive editor of GMD, I would like to bring to your attention our Editorial version 1.1:

http://www.geosci-model-dev.net/8/3487/2015/gmd-8-3487-2015.html

This highlights some requirements of papers published in GMD, which is also available on the GMD website in the 'Manuscript Types' section:

http://www.geoscientific-model-development.net/submission/manuscript_types.html

In particular, please note that for your paper, the following requirements have not been met in the Discussions paper:

- "The main paper must give the model name and version number (or other unique identifier) in the title."

- "If the model development relates to a single model then the model name and the version number must be included in the title of the paper. If the main intention of an article is to make a general (i.e. model independent) statement about the usefulness of a new development, but the usefulness is shown with the help of one specific model, the model name and version number must be stated in the title. The title could have a form such as, "Title outlining amazing generic advance: a case study with Model XXX (version Y)"."

In order to simplify reference to your developments, please add a model name and a version number in the title of your article in your revised submission to GMD.

Yours,

Astrid Kerkweg

---

## Author Comment (AC1) · 11 Aug 2016

Dear Astrid Kerkweg,

Thank you for your comment. I have now updated the title and added a model name and version number. The new title of the manuscript is:

Application of the adjoint approach to optimise the initial conditions of a turbidity current (AdjointTurbidity 1.0)

I also changed the name of the code repository to reflect the new model name:

https://bitbucket.org/simon_funke/adjoint-turbidity

Best wishes,

[Figure]

Simon

---

## Referee Comment (RC1) · Anonymous Referee #1 · 6 Oct 2016

**1   General remarks**

This paper addresses the problem to find optimal turbidity current parameters in the initial condition that reproduce sediment deposits in real world through computer simulation. Solving this interesting inverse problem would improve the interpretation of sedimentary successions. A gradient-based optimization method has been used to minimizes the misfit between modeled and observed field sediment deposits. The gradient calculation, which is the most challenging part, is obtained using the adjoint approach with the adjoint model being automatically generated by the high-level automatic differentiation tool dolfin-adjoint. Although the solution found cannot make the modeled deposit exactly match the measured deposit, the paper demonstrates the power of gradient-based optimization and the automatic tools based on high-level ab-

straction such as dolfin-adjoint and FEniCS. It is well written and easy to follow. The main weakness is the poor solution found by the optimization, but it can be improved, or at least compensated. Considering the novelty of the work, I think this paper merits publication in GMD after minor revisions.

1. The objective function can be considerably improved. Figure 5 shows that the Bed 1.1 profile has a depth around 0.5 cm at the right boarder. But in the simulated results the deposit depth is always zero at the right boundary (y=1) according to (17). This can also be noticed from Figure 7. So the target profile used in the objective function (53) does not represent a complete profile, while the simulation always provides a complete one. One possible way for a better comparison is to extrapolate the target profile further to the right until the depth becomes zero, and then define the functional on the range $[0, \hat{x}_N]$ instead of $[0, \hat{x}_{max}]$

$$J = \int_0^{\hat{x}_N} (\tilde{\eta} - \eta^T)^2 d(\hat{x}). \tag{1}$$

This will make the calculation easier and avoid the difficulty with discontinuities. Originally discontinuity may arise when $\hat{x}_N > \hat{x}_{max}$ because the model profile is truncated at $\hat{x}_{max}$ in the integral evaluation and then the functional depends on only part of the system states. Furthermore, there would be no need to use the scaled filers and the nasty approximation of max and min functions.

2. It may be worth trying different end times for the simulation. The optimal modeled profile found in this work looks much shorter than it should be compared to the target profile, leaving a significant gap towards the end. An ideal model profile that matches well with the observation would be expected to have longer length than the extent of measured data ($\hat{x}_N > \hat{x}_{max}$). Is it possible to get a longer profile by increasing the end time $t_f$? If so, the quality of the solution would be improved.

3. More evidence should be provided to support the claim that the gradient-based optimization methods are powerful for solving the inverse problem of find turbidity current parameters. Why not generate a reference profile with a known initial condition and run the same code with different initial guesses to see if the reference profile can be reproduced in the optimal solution? This artificial test will give more insight on the performance of the optimization techniques. And hopefully this will also make it clear if there are local minimums that lead to low-quality solutions.

4. There are not enough instructions on how to run the code. For example, FEniCS is suggested to be run with Docker, however, it is not clear how to get all the components work with Docker, or without Docker. The specific commit of libadjoint described on the bitbucket page does not compile with gfortran 5.3 on my mac, but the latest commit compiles without any problem. It would be beneficial for the readers who want to play with the code if the workflow could be simplified (e.g. wrap everything into a FEniCS Docker image) and described in more detail. Based on the current available resource, I am not able to confirm the reproducibility of the results, although I have no doubt with them.

**2  Specific remarks**

1. Some entries in the references use abbreviations for journal names but some others use full names. Whatever format is preferred, it should be consistent.

2. $\eta T$ should be $\eta^T$.

3. Line 26: "spent in" should be "spent on".

4. Line 28: what does "similarity solution" mean? Is it "similar solution"?

5. Line 323: "that improve on" should be "to improve".

6. Line 450: "$max(\hat{x}_N, \hat{x}_{max})$" should be "$min(\hat{x}_N, \hat{x}_{max})$".

7. ˜is misplaced in the caption on (e) in Figure 11.

8. Units are missing in Figures 7, 9 and 10.

9. Line 643: "the addition of parameterisations" should be "additional parameters".
* * *

---

## Referee Comment (RC2) · Anonymous Referee #2 · 17 Nov 2016

This paper by Parkinson provides a great insight into the primary drivers of the sediment transport process from continental shelf to deep ocean. The choice of the shallow water sediment-laden density current model and its numerical solution using finite element mixed DG method with adaptive time-stepping are quite appropriate. The crucial features of this paper that make it original and unique are the use of adjoint based assimilation to derive optimal initial parameters that best fit the field measurements, and its application to a real world problem. The paper is very well organized and the mathematical equations are clearly stated which make it easy to follow. The paper deserves publication in the journal of Geoscientific Model Development with few major and minor corrections stated as follows

**Major**:

1. In line 69-71, the authors claim that the paper represents first published work with optimization applied to turbidity currents. However, this needs to be reworded to state that this is first published work of application of adjoint based optimization applied to turbidity currents demonstrated through real world example. The reason is that the article *"Towards inverse modeling of turbidity currents: The inverse lock-exchange problem"* published in Computers and Geosciences, authored by Lutz Lesshafft et. al., 2011 ([http://www.off-ladhyx.polytechnique.fr/people/lutz/pdfs/Lesshafft_CAGEO_2011.pdf](http://www.off-ladhyx.polytechnique.fr/people/lutz/pdfs/Lesshafft_CAGEO_2011.pdf)) provides a similar approach, however, their approach is gradient-free optimization. It is also suggested to include reference to this article.

2. As seen in sections 4.4 and 4.6, the model based deposit profile does not match completely the target profile. The authors are suggested to perform the following tests to assess their setup

    a. For every adjoint model code produced through automatic differentiation or manually, it is important to validate it. The authors have provided verification of gradient calculations in section 4.3, however, it would be useful to perform box model testing and generate scatter plots similar to the plots presented in section 1.1.5 of the article "*Towards the construction of a standard adjoint GEOS-Chem model*", Proceedings of the 2009 Spring Simulation Multiconference (paper draft: [http://people.cs.vt.edu/asandu/Deposit/draft_2009_gc-adj.pdf](http://people.cs.vt.edu/asandu/Deposit/draft_2009_gc-adj.pdf)). This would ensure that the adjoint code produced by dolphin-adjoint tool is correct.

    b. Although this step is not mandatory for publication of this article, it would also be useful to set up a reference profile and conduct an experiment similar to Lesshafft et. al. to test if the implementation of all the numerical methods solving the shallow water model and the adjoint are in fact working correctly, before applying it to the deposit profile of Marnoso Arenacea Formation.

**Minor**:

1. Line 83: "which takes account of the" -> "which takes into account the"

2. Line 84: "retarding" -> "impeding"

3. $\eta^T$, $\eta T$ and $\eta_T$ have been used interchangeably in figures 7 and 9, and in Section 4.2 onwards

4. unit missing in height ($h_0$) in the optimized values on line 576 and input parameter on line 593, should be meters

---

## Author Comment (AC2) · 5 Jan 2017

We would like to thank the reviewer for her/his constructive comments.

Below, we provide answers for each of the reviewer's comments:

> 1.The objective function can be considerably improved. Figure 5 shows that the > Bed 1.1 profile has a depth around 0.5 cm at the right boarder. But in the simulated > results the deposit depth is always zero at the right boundary (y=1) according > to (17). This can also be noticed from Figure 7. So the target profile used in > the objective function (53) does not represent a complete profile, while the simulation > always provides a complete one.

The simulated profile is indeed always zero on the right boundary. However, the model

allows for profiles that go beyond the observed target profile. Hence the simulated profile may take non-zero values where the target profile stops, and the functional allows a match of the simulated and target profiles.

> One possible way for a better comparison > is to extrapolate the target profile further to the right until the depth becomes zero, > and then define the functional on the range [0, xĚĘN ] instead of [0, xĚĘmax]

We see two problems with this approach:

1. Extrapolation is generally a "dangerous" process. For example, if one only had a very small target profile, maybe over 1 km, one would be faced with extrapolating this profile to xˆN - and depending on the extrapolation strategy one would probably obtain different reconstructions.

2. The simulation end time, and the length of the simulated profile xˆN, are not known a priori. More precisely, the simulation is terminated once all sediment in the fluid has is settled to the sediment, which also means that the simulated profile does not change further after this point.

> This will make the calculation easier and avoid the difficulty with discontinuities. > Originally discontinuity may arise when xĚĘN > xĚĘmax because the model profile is > truncated at xĚĘmax in the integral evaluation and then the functional depends on > only part of the system states. Furthermore, there would be no need to use the > scaled filers and the nasty approximation of max and min functions.

We agree that it would be nice to simplify the functional - but we do believe that our functional yields a more robust reconstruction, even if it means that we have to handle the additional complexity.

> 2. It may be worth trying different end times for the simulation. The optimal modeled > profile found in this work looks much shorter than it should be compared to the > target profile, leaving a significant gap towards the end. An ideal model profile > that

matches well with the observation would be expected to have longer length > than the extent of measured data ($x\hat{}N > x\hat{}max$). Is it possible to get a longer > profile by increasing the end time tf? If so, the quality of the solution would be > improved

Extending the simulation would not change the length or the shape of the profile further. The reason is that the simulation is terminated only after nearly all particles are settled into the sediment. This means that further simulation time will not have significant impact on the sediment profile. The conclusion of the manuscript discusses a number of other reasons why the optimised profile does not match the target profiles more precisely - most likely our numerical model is missing some important physical details, which could be an interesting future direction of this research.

> 3. More evidence should be provided to support the claim that the gradient-based > optimization methods are powerful for solving the inverse problem of find turbidity > current parameters. Why not generate a reference profile with a known initial condition > and run the same code with different initial guesses to see if the reference > profile can be reproduced in the optimal solution? This artificial test will give more > insight on the performance of the optimization techniques. And hopefully this will > also make it clear if there are local minimums that lead to low-quality solutions.

We appreciate your comment, and it is a very good point. However, we feel that this is already covered in the paper as it stands now. Although this very simple model could feasibly be optimised manually, a more complex, and complete model would be very difficult to optimise by manually tweaking the input parameters. Specifically lines 315-322 cover this, and at numerous locations in the paper we discuss how improvements to the model are required to better model the flow - many of which would greatly increase the number of input parameters leading to big efficiency gains by using a gradient-based optimisation.

> 4. There are not enough instructions on how to run the code. For example, FEniCS > is suggested to be run with Docker, however, it is not clear how to get all > the
components work with Docker, or without Docker. The specific commit of > libadjoint described on the bitbucket page does not compile with gfortran 5.3 on > my mac, but the latest commit compiles without any problem. It would be beneficial > for the readers who want to play with the code if the workflow could be > simplified (e.g. wrap everything into a FEniCS Docker image) and described in > more detail. Based on the current available resource, I am not able to confirm the > reproducibility of the results, although I have no doubt with them.

Installation is indeed a problem, and the FEniCS Docker idea is great. We have now created a Docker file that should simplify the installation significantly. The details are describe on the official bitbucket webpage: https://bitbucket.org/simon_funke/adjoint-turbidity/overview

Specific comments ———————— All specific comments have been addressed. In particular, we now to use \eta_T rather than \eta^T throughout the paper (it was inconsistent before). We decided on this notation to be in line with \eta_0 (the initial sediment profile).

Please also note the supplement to this comment:
http://www.geosci-model-dev-discuss.net/gmd-2016-136/gmd-2016-136-AC2-supplement.pdf

---

## Author Comment (AC3) · 5 Jan 2017

We would like to thank the reviewer for her/his constructive comments.

Below, we provide answers for each of the reviewer's comments:

» Major: » 1. [...] "Towards inverse modeling of turbidity currents: The inverse lock-exchange problem" published » in Computers and Geosciences, authored by Lutz Lesshafft et. al., 2011 (http://www.offladhyx.polytechnique.fr/people/lutz/pdfs/Lesshafft_CAGEO_2011.pdf) » provides a similar » approach, however, their approach is gradient-free optimization. It is also suggested to include » reference to this article.

The reference is very relevant and has been added to the manuscript.

» 2a. [...] and generate scatter plots similar to the plots presented in section 1.1.5 of the article » "Towards the construction of a standard adjoint GEOS-Chem model", Proceedings of the » 2009 Spring Simulation Multiconference (paper draft: » http://people.cs.vt.edu/asandu/Deposit/draft_2009_gc-adj.pdf). This would ensure that » the adjoint code produced by dolphin-adjoint tool is correct.

We have extended section 4.3 by explicitly stating the Taylor test. In addition, we added table 1, which shows the Taylor remainders and the convergence order (similar to the mentioned scatter plot) for a specific example.

» 2b. Although this step is not mandatory for publication of this article, it would also be useful » to set up a reference profile and conduct an experiment similar to Lesshafft et. al. to » test if the implementation of all the numerical methods solving the shallow water model » and the adjoint are in fact working correctly, before applying it to the deposit profile of » Marnoso Arenacea Formation We have performed such tests successfully during the code development. For the interested reader, the code for these tests is submitted in the code repository https://bitbucket.org/simon_funke/adjoint-turbidity/src/master/tests/optimise_scalar_params

Specific comments —————— All specific comments have been addressed. In particular, we now to use \eta_T rather than \eta^T throughout the paper.

Please also note the supplement to this comment:
http://www.geosci-model-dev-discuss.net/gmd-2016-136/gmd-2016-136-AC3-supplement.pdf

[Figure]

**Supplement:**

[revised manuscript text omitted]